# FEP Encapsulated Crack-Based Sensor for Measurement in Moisture-Laden Environment

**DOI:** 10.3390/ma12091516

**Published:** 2019-05-09

**Authors:** Minho Kim, Hyesu Choi, Taewi Kim, Insic Hong, Yeonwook Roh, Jieun Park, SungChul Seo, Seungyong Han, Je-sung Koh, Daeshik Kang

**Affiliations:** 1Department of Mechanical Engineering, Ajou University, Suwon 16499, Korea; hgh7706@ajou.ac.kr (M.K.); michelle0407@ajou.ac.kr (H.C.); rsb5287@gmail.com (T.K.); his3080@gmail.com (I.H.); ckj2065@naver.com (Y.R.); n9near@ajou.ac.kr (J.P.); 2Department of Health, Environmental and safety, EulJi University, Seongnam 13135, Korea; seo@eulji.ac.kr

**Keywords:** strain sensor, thin metal film, spider inspired, crack, encapsulation

## Abstract

Among many flexible mechanosensors, a crack-based sensor inspired by a spider’s slit organ has received considerable attention due to its great sensitivity compared to previous strain sensors. The sensor’s limitation, however, lies on its vulnerability to stress concentration and the metal layers’ delamination. To address this issue of vulnerability, we used fluorinated ethylene propylene (FEP) as an encapsulation layer on both sides of the sensor. The excellent waterproof and chemical resistance capability of FEP may effectively protect the sensor from damage in water and chemicals while improving the durability against friction.

## 1. Introduction

In an endeavor to minimize the size of electronics and develop wearable devices, flexible, stretchable, and ultra-thin sensors such as piezoelectric and piezoresistive strain sensors [1,2,3,4,5,6] have emerged in various scientific fields [7,8,9,10,11,12,13,14,15,16,17]. A spider-inspired crack-based sensor is one of these flexible electronics that has been receiving attention due to its filmy, light, and highly sensitive characteristics [18]. The crack-based sensor can be easily produced and mounted on human skin. And, inspired by a spider’s slit organ, the sensor has cracks on its surface. In terms of the crack-based sensor’s working principle, when the sensor is extended in the axial direction, the cracked film compressed in the transverse direction due to Poisson’s ratio. Since there are numerous sizes of crack asperity, some prominent crack edges reconnect to the opposite prominent crack edges due to the lateral compression. As the sensors gain larger strain, the number of contacts decreases. The reduction of the number of contacts results in an increase in the sensor’s resistance. As a result, the cracks’ disconnection-reconnection behavior yields high sensitivity over 2000 gauge factor (GF) at 2% strain. Gauge factor (GF), a significant factor of the sensor’s sensitivity, is defined as (ΔR/R_0_)/ε where ΔR is the difference of resistance, R_0_ is initial resistance and ε is deformation rate. Compared to other sensors which have comparable GF, the crack-based sensor not only has higher stretchability but also can be easily fabricated [19].

In spite of these great advantages, the crack-based sensor needs further development due to its vulnerability in terms of over-cracking, delamination, and wetting. Recently, there have been many attempts to enhance the crack-based sensors’ performance [20,21,22,23] and durability [24,25] by altering the materials or the fabrication. However, there was not an innovative approach that can enhance durability and protect the sensor from water, chemicals, friction, dust and short circuit at the same time. Here, we have adopted the encapsulation method to improve the sensor’s durability for wider applications, including implantable devices, because it can protect a subject from oxidation, friction, and chemicals [26,27,28,29,30,31,32]. Recently, Kim et al. encapsulated the crack-based sensor with polyimide (PI) film [24]. However, the PI encapsulated crack-based sensor was able to endure an underwater environment only for a few hours. To develop the performance of the crack-based sensor, We encapsulated the crack-based sensor with fluorinated ethylene propylene (FEP) as a material to enhance the sensor’s mechanical, water, and chemical resistance by taking advantage of its low water absorption (0.010%) and high chemical resistance [33,34,35,36,37,38]. FEP’s unique characteristics such as low water vapor transmission rate (WVTR) and high chemical resistance is due to the strong bonding between carbon atom chains and fluorine atoms. Whereas the polyimide film with a 25 µm thickness has a WVTR of 54 g/(m^2^·24 h) (Kapton Type 100 HN film, DuPont, Wilmington, DE, USA), FEP film with same thickness has a WVTR of 4.65 g/(m^2^·24 h). The encapsulation process can be easily conducted by pressing FEP films at high temperature. In addition, we used a polyimide (PI) film as a substrate to cope with high temperature and harsh mechanical damage [39,40]. In order to prove its adequacy, we conducted several experiments such as water permeability test, chemical resistance test, and durability test. We anticipate that this advanced sensor could be actively used on human bodies and even an underwater environment.

## 2. Experimental Section

### 2.1. The Configuration and Fabrication of an FEP Encapsulated Crack-Based Sensor

Figure 1 shows the overall configuration of the FEP encapsulated crack-based sensor. The physical size of the sensor is 10 mm in width, 68 mm in length, and 35 mm in gauge length. Structurally, it is composed of five layers as shown in Figure 1a. For the substrate, a 7.5 µm thick PI film (3022-5 Kapton thin film, Chemplex, Palm City, FL, USA) was employed to endure the encapsulation process that is performed in high temperature (280 °C). Prior to the deposition process of metal layers, plasma treatment was performed by an oxygen plasma system (CUTE, Femto science Co., Hwaseong-si, Gyeonggi-do, Korea) at 100 W for 10 min, 0.5 torr pressure, and 30 sccm flow rate in order to enhance the adhesion between the PI film and the metal layers [41]. 50 nm thick chromium (Cr) and 30 nm thick gold (Au) were deposited in sequence by a thermal evaporation system (Thermal Evaporation System, DD High Tech. Co., Gimpo-si, Gyeonggi-do, Korea) to function as a crack inducing layer and an electrical conductor, respectively. Two 25 µm thick FEP films (BB3090-1-24, Lake Havasu City, AZ, USA) enclosed the sensor to protect it from friction, water, and chemical damages. Figure 1b is a scanning electron microscope (SEM) image of the cross-section of the FEP encapsulated sensor that shows the layers pseudo-colored for distinction. The encapsulation process was implemented by pressing the sensor placed between the FEP films at 50 bar pressure with a heating press that is heated to 280 °C (Figure 1c). After this process, cracks were created by using a material testing machine (3342 UTM, Instron Co., Norwood, MA, USA) with 0–2% strain, 20 mm/min speed, and 2000 cycles (Figure 1d). The chromium layer acts as a crack inducing layer so that the cracks occur on the metal layers by stretching the sensor. To reduce errors and stabilize crack creation, we set standards such as initial length, strain range, tension speed and cycles. Appendix A shows the reproducibility of the FEP encapsulated crack-based sensor obtained from five different sensors. We observed a cross-section of the sensor using a focused ion beam (FIB) to demonstrate the size of the crack. The size of the crack was 100–200 nm (Appendix A).

### 2.2. Method and Test

To verify that the encapsulation enhances durability, water, and chemical resistance of a crack-based sensor, the FEP encapsulated sensor has completed durability test, underwater test, chemical resistance test, and thermal resistance test. The water vapor transmission rate (WVRT) of the FEP films was measured by conducting an ASTM F1249 test with PERMATRAN-W^®^ Model 3/33 (Mocon Inc., Brooklyn Park, MN, USA) which is known as a standard test method for WVTR through plastic film and sheeting using a pressure modulated sensor. The test method was to seal FEP film between the wet chamber and dry chamber for about 13 h measuring moisture transmitted through the FEP film using the pressure modulated sensor. It was done at a selected temperature (38 °C) and humidity (100% relative humidity) to avoid uncertainness. The underwater test has been carried out by immersing the sensor in deionized (DI) water for 18 days and measuring resistance every 3 days. In the chemical resistance test, the sensor has been soaked in chromium etchant for 14 days and measured every 2 days. At last, the thermal resistance test was implemented by placing the sensor on a hot plate for 10 min from room temperature (25 °C) to 75 °C and measured at every 10 degrees Celsius. The resistance variation of the sensor was measured by using a material testing machine (3342 UTM, Instron Co., Norwood, MA, USA) with 0–2% strain, 40 mm/min speed, and 40 cycles. To measure the resistance of sensors in a stable manner, we applied conductive epoxy to connect wires on the sensor and also used NOA68 (Norland Optical Adhesive 68, Norland Products Inc., Cranbury, NJ, USA), known for improving adhesion to many plastic films, on conductive epoxy to fix it reliably (Appendix A). All the data was collected by a LabVIEW based data acquisition system (PXI-4071, National Instruments Inc. Austin, TX, USA). The standard deviation of the measurement was 0.01 Ω measuring 2 wire resistance with 5 1/2 digit.

## 3. Results and Discussion

### 3.1. Basic Electrical Characteristics of FEP-Encapsulated Sensor

It is one of the most essential requisites for flexible, stretchable thin film sensors to maintain the performance against repeated deformation. Due to the stress concentration, the crack of previous crack-based sensors is easily deepened which leads to the degradation of the sensors. A moisture-laden environment also causes delamination, corrosion, and oxidation to the sensors.

Particularly, to be used as wearable or medical devices, the sensors need to endure human secretion such as perspiration, saliva or gastric juice. To ensure this use, we encapsulated the crack-based sensor with FEP films which are known for their high resistance against water and chemicals as well as biocompatibility.

We conducted three basic experiments such as hysteresis test, normalized resistance test, and durability test to demonstrate the electrical characteristics of the FEP-encapsulated sensor. Figure 2a is the result of the hysteresis test that shows the resistance variation depending on strains when the sensor gains loading and unloading force, and the shapes of graphs are nearly identical. The reason for the subtle hysteresis error is that the PI substrate has a high Young’s modulus and low stretchability so the sensor can return to the initial state rapidly. Even though the sensor response is nonlinear, it is in one-to-one correspondence. So with the aid of calibration, the sensor can be used as a strain sensor properly. We repeatedly stretched the sensor with 0–2% strain and 40 mm/s velocity. The peaks of resistance are consistent (Figure 2b).

In the durability test, five different sensors endured 0–2% strain repeated up to 15,000 cycles After 15,000 cycles, the average of sensors’ GF remained above 84% (Figure 2c). Since FEP has lower Young’s modulus than PI, FEP films gain lower stress than PI films in the elastic deformation region. Therefore, when they stick together, the sensor has a lower stress concentration on their cracks so the encapsulated sensor has higher durability than a bare crack-based sensor. In addition, we carried out a test that shows the difference of resistance varied with the thickness of FEP film. The non-encapsulated crack sensor has the highest resistance, and the resistance decreases as the thickness of FEP film increases (Figure 2d). Although encapsulated sensors have a much lower GF than non-encapsulated sensors, the GF of encapsulated sensors is sufficient to detect a slight change of mechanical deformation. Additionally, the GF comparison of the crack-based sensors is in Appendix A to ensure the length of the sensor is not a crucial factor of GF.

### 3.2. Water Permeability and Chemical Resistance of the FEP-Encapsulated Sensor

Figure 3a shows the water vapor transmission rate (WVTR) of 25 µm and 50 µm thick FEP films, and the data were gathered by Korea Polymer Testing and Research Institute (Koptri). The WVTR of the 25 µm thick FEP film is about 4.82 g·m^2^/day on average and the 50 µm thick FEP film is about 2.04 g·m^2^/day on average. It means that the sensor encapsulated with the 50 µm thick FEP film endures the underwater environment longer than the sensor with the 25 µm FEP film. To be convinced with the water-resistant ability of the FEP encapsulated sensor, two sensors encapsulated each with the 25 µm and 50 µm thick, FEP films have been submerged in water for 18 days and measured resistance every 3 days (Figure 3b). The sensor with the 25 µm thick FEP film has remained above 98% of GF compared to the initial state until the 3rd day and has fallen to 80% of initial GF at 6th day. The sensor’s GF has been continuously decreased due to the rise of the base resistance and ended up below 38% GF on the 18th day (Appendix A). The sensor with the 50 µm thick FEP film kept above 93% of initial GF until the 6th day which has decreased to 72% on the 18th day. Additionally, a non-encapsulated crack-based sensor was destroyed within 32 h in water (Appendix A).

In the chemical resistance test, the sensor encapsulated with the 25 µm FEP film has been immersed in chromium etchant for 14 days and resistance was measured every two days (Figure 3c). The GF has reserved above 90% GF until the 4th day. From the 10th day, the GF decreased dramatically resulting 23% GF on the 14th day (Appendix A). We also measured the resistance variation of 50 µm FEP encapsulated sensor in chromium etchant for 14 days. unlike the sensor with 25 µm FEP film, the 50 µm FEP encapsulated sensor was not destroyed on the 14th day. Interestingly, time goes by, while the shape of the graph of the underwater test was maintained, that of the chemical resistance test was degraded (Figure 3b,c). This is because chromium etchant dissolves immediately the Cr layer in the crack-based sensor while water takes a long time to dissolve the metal layers.

To demonstrate that the sensor performs properly in human body temperature and even in higher temperature, resistance variation of five different sensors with the 25 µm FEP film was measured from room temperature (25 °C) to 75 °C (Figure 3d). The GF has been maintained about 90% compared to the beginning until 55 °C and has fallen to 82% GF at 75 °C (Appendix A).

### 3.3. Underwater Motion Test

Along with the development of flexible and stretchable electronics, recent medical devices attempt to focus on light, thin, and skin-mountable characteristics to gain real-time data. Our sensor has great potential to be applied to human skin and medical devices because it is simply usable, easily producible, bio-compatible, and waterproof. The sensor can be easily mounted on a curved surface, such as a human finger, due to its flexibility and thinness (Figure 4a). We attached the sensor in the direction of the radius of a finger since when the finger bends or spreads, the diameter of the finger changes. To demonstrate that the sensor can measure motion in water, we measured the sensor’s resistance variation while the finger with the sensor mounted on crooked in air and water (Figure 4b). Figure 4c shows the resistance variation graph when the sensor mounted on the finger gained strain from the finger’s motion. The left side of the graph shows the measurement in air and the right side in water. To convince the sensor response in air and water is statistically significant, we implemented two sample t-test. We calculated mean and standard deviation of every peak in the data and Appendix A shows the results. As the result of a two sample t-test, the t statistic (t) was 6.3930, the degree of freedom was 18 and the two-tailed *p*-value was less than 0.0001 that means the difference between the data is considered to be extremely statistically significant. The influence of wire shaking in the test is negligible compared to the resistance variation of finger motion (Appendix A). Furthermore, to prove that the sensor can accurately gain vital signs in water, we measured human pulse rate in water with the sensor encapsulated with the 25 µm thick FEP film (Figure 4d). The gathered signals were stable and precise even though the pulse signal is very weak (Figure 4e). In addition, to show the sensor can measure finger motion and human pulse rate in water for a long time, we implemented long term underwater test and the results are shown in Appendix A. Although the output resistance of the sensors are slightly different because the measuring environment is different every day, the shape of the signals was clear until the 9th day.

## 4. Conclusions

We have demonstrated that the crack-based sensor with FEP encapsulation has many advantages. The FEP encapsulation of crack-based sensor provides stronger durability, water and chemical resistance than a non-encapsulated sensor. To prove these advantages of FEP encapsulation, we conducted a durability test, an underwater test, and chemical test. In the durability test, the FEP encapsulated sensor endured over 15,000 cycles of 0–2% strain. In underwater test, the sensors were immersed in water for 18 days. The sensor encapsulated with the 25 µm FEP film kept 98% of initial GF until the 3rd day while the sensor with the 50 µm FEP film remained 93% of initial GF until the 6th day. When the sensor with the 25 µm FEP film has been submerged in chromium etchant for 14 days, the sensor’s sensitivity remained 90% of initial GF until the 4th day. As a result, in terms of the durability, water and chemical resistance, the FEP encapsulated sensor has great potential to be utilized in wearable devices or medical devices that should endure human secretion such as perspiration, saliva, or gastric juice.

## Figures and Tables

**Figure 1 materials-12-01516-f001:**
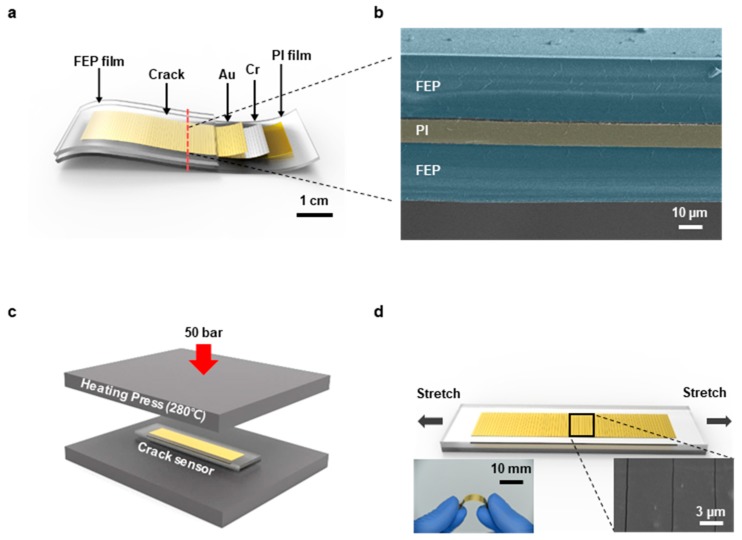
(**a**) Schematic illustration of the fluorinated ethylene propylene (FEP) encapsulated sensor’s composition. (**b**) Scanning electron microscope (SEM) image of the sensor’s cross section. (**c**) Schematic illustration of the encapsulation process conducted by a heating press. (**d**) Schematic illustration of the crack-forming process and a photograph of the FEP encapsulated sensor and an SEM image of cracks on the sensor.

**Figure 2 materials-12-01516-f002:**
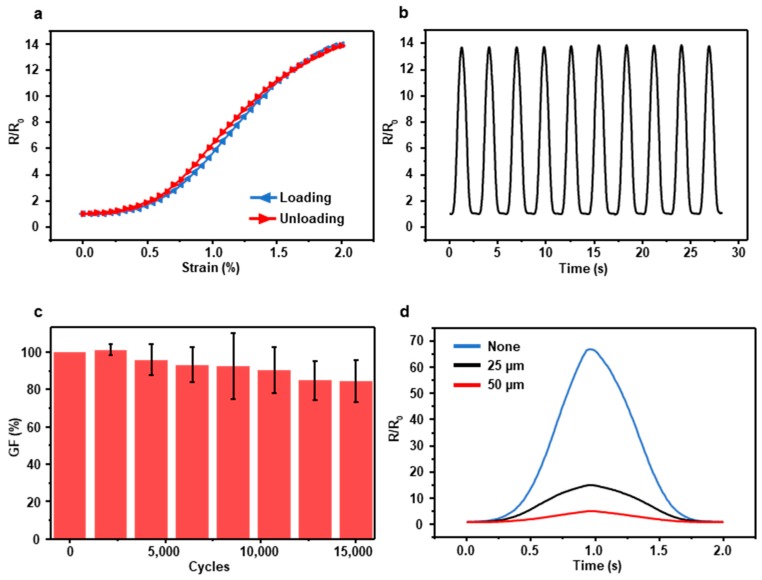
(**a**) Hysteresis curve of the sensor encapsulated with the 25 µm FEP film when the sensor receives loading and unloading force. (**b**) Normalized resistance at 0–2% strain and 40 nm/s velocity. (**c**) The durability of the sensor encapsulated with the 25 µm FEP film. (**d**) Resistance variation depending on the thickness of the FEP film.

**Figure 3 materials-12-01516-f003:**
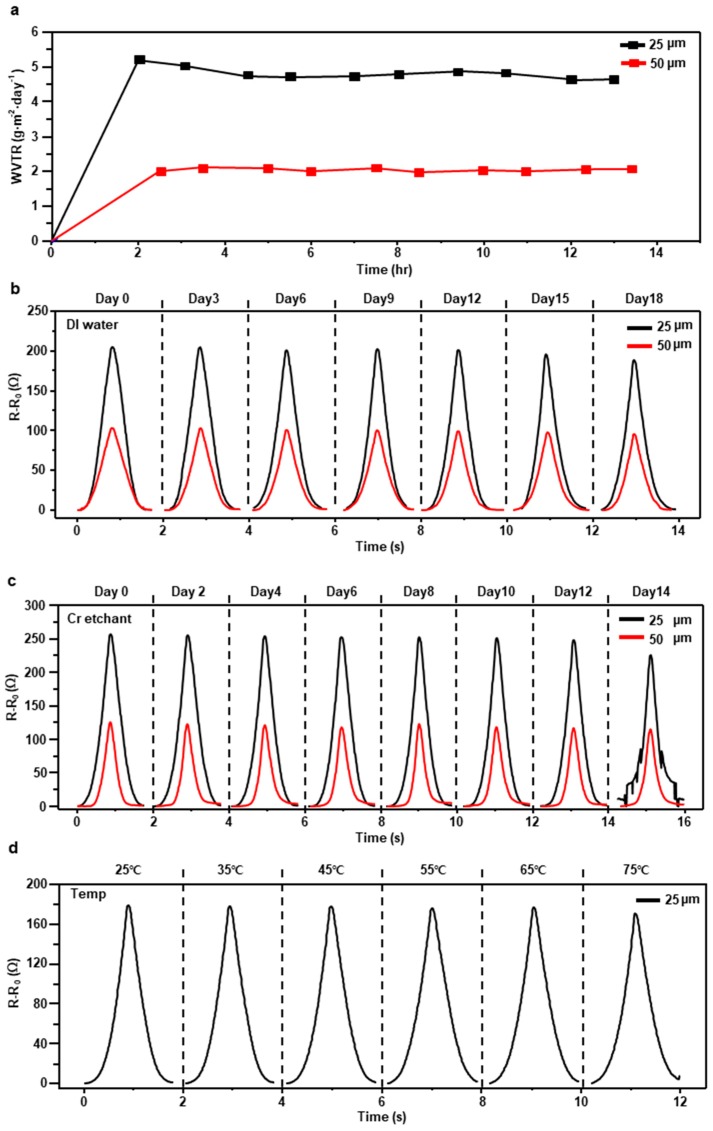
(**a**) Water vapor transmission rate (WVTR) of the 25 µm and 50 µm thick FEP films. (**b**) Resistance variation of sensors encapsulated with the 25 µm FEP film and 50 µm FEP film that have been soaked in water for 18 days. (**c**) Resistance variation of the sensor encapsulated with the 25 µm FEP film that has been immersed in chromium etchant for 14 days. (**d**) Resistance variation of the sensor encapsulated with the 25 µm FEP film from 25 °C to 75 °C.

**Figure 4 materials-12-01516-f004:**
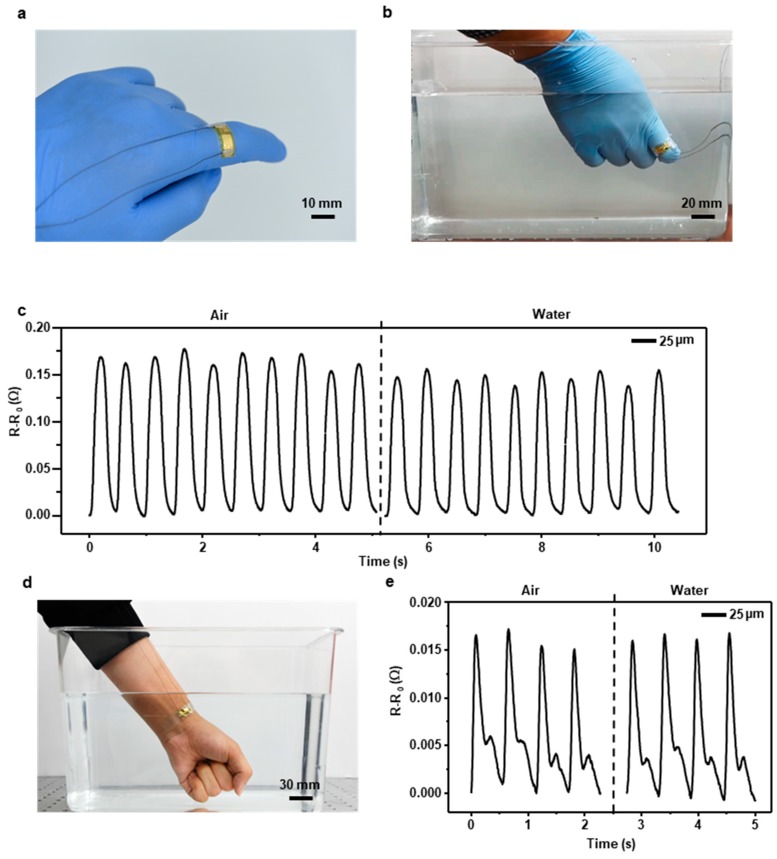
(**a**) Overall image of the FEP encapsulated sensor mounted on a finger. (**b**) Underwater motion test to measure the actions of the finger. (**c**) Performance of the sensor that measured the finger’s motions in air and water. (**d**) Underwater vital sign test to measure pulse rate in water. (**e**) Performance of the sensor that measured pulse rate in air and water.

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
