# Peer review of "FEP Encapsulated Crack-Based Sensor for Measurement in Moisture-Laden Environment"

_materials, 2019, doi:10.3390/ma12091516_

Round 1
Reviewer 1 Report
Strain sensors are highly interesting and applicable in different fields. In this paper, authors presented fabrication process and testing a cracked-based strain sensor. Results presented are not discussed well and the main contribution is not clear. Please note that just listing the results is not enough, you should analyse each and every plot presented. Please refer to my below comments for further details.
1) Introduction section needs to be improved. It is important to compare the other sensing principle such as piezoelectric (such as the below paper) and piezoresistive strain sensors with crack-based sensor
H. Khan, A. Razmjou, M. Ebrahimi Warkiani, A. Kottapalli, M. Asadnia, Sensitive and Flexible Polymeric Strain Sensor for Accurate Human Motion Monitoring, Sensors, 18 (2018) 418
2) What is the novelty of the sensor? Crack-based sensor with the fabrication process proposed has been around for many years. Authors need to clarify their novelty and contribution.
3) Fabrication process needs more details especially in the process of crack creation. How reliable is the process? What is the reproducibility of the device? What is the size of the crack?
4) I do not see the working principle of the sensor anywhere in the text. Basically, the author needs to explain why this sensor function. How the crack is used to sense strain.
5) Hysteresis error presented in figure 2 is very low which is interesting. Authors need to analyse the results and explain why this error is lower than that of piezoelectric sensors.
6) How did you measure the resistance? Experimental setup for data acquisition has not been mentioned in the text.
7) Water permeability and chemical resistance are due to FEP film which has been used for the encapsulation. More interesting experiments could be the effect of the size of the sensor. Authors need to comment on how would the sensor output change when the size of the sensor is smaller or bigger.
8) I would imagine a high wire shaking occurs during all the oscillation experiments in figure 3 and 4. The wire shaking itself could give false results. What precautions has been considered to ensure the output presented is actually coming from the sensor and not other disturbances?
9) How many sensors have you tested in total? Authors need to perform the experiments for at least 5 sensors and plot the error bars to ensure the results are repeatable. Often the issue with crack-based sensors is that since the crack creation is a random process every two sensors would behave slightly differently which limits their practical application.
Author Response
We thank you for your valuable comment. We added sentences that mention comparison with other sensors in the main text.

Reviewer 2 Report
This manuscript describes a crack-based sensor with FEP encapsulation providing strong durability over 15,000 cycles of 0-2 % strain, immersed in water for 18 days and chemical resistance by submerged in chromium etchant for 14 days. Some demonstration was showed for use of the sensor in wearable devices or medical devices, including the measurement of human motion and pulse rate. Overall, the manuscript is in a good shape ad the development of flexible sensors toward reliability and durability is of interests for wide range of wearable applications. However, considerable revisions should be made before publications. My detailed comments are as follows.
1.In material engineering point of view, are there any advantages of using Fluorinated Ethylene Propylene for encapsulation compared with Polyimide? This should clarify in the introduction.
2. What is the advance of this work compared to the very similar work (DOI 10.3390/app8030367)? In addition, benchmarking with the literature should be provided to show the progress in materials engineering for crack-based strain sensors.
3. The authors should provide SEM/TEM images showing the crack mechanism as it is the core concept of the sensor. How good is the bonding between PI and FEP under water/chemical? discussion on the impact of chemical on thickness change of FEP could be mentioned
4. How many samples did the authors test? And what is the reproducibility of these sensors?
5. It is not clear how to make interconnections for testing? Is the contact resistance between sensor/electrode change with chemical/water? Discussion should be given.
6. Figure 2a indicates that the sensor response is non linear with applied strain. How this sensor can be used for practical applications where accurate strain must be determined?
7. The sensor is thin and strain detection limits at only 2%. How this sensor can be suitable for finger motion detection and other applications required stretchable capability? Please discuss.
Author Response
We thank you for your valuable comment. We added sentences that mention comparison with Polyimide encapsulation.

Reviewer 3 Report
Interesting work which could be published after major revision .
The authors cover their crack-based resistive sensors with FEP films, and perform
long-time tests on their performance.
1) I think it would be good if the authors describe their experimental procedures in some more detail: How was the water vapor transmission rate (WVTR) in Fig. 3 a) measured? How does the WVTR-values compare with those reported in the literature? Since FEP in itself does not allow any water vapor to pass (water is too large), the water molecules must diffuse into the edges of the measurement setup. If not, please explain, and provide evidence from the literature that water vapor actually transmits through FEP. I have worked a lot with FEP isolating sensitive electrical equipment, and not experienced noticeable water transmission. It is therefore my opinion that the authors should explain their results and compare with existing literature.
2) The authors should conduct long-time underwater tests (Fig. 4), corresponding to those in Fig. 3. Otherwise it is hard to understand the logics in doing the long term tests in Fig. 3. Moreover, the authors should explain in more detail whether there are any statistically significant differences between air and water in Fig. 3 c).
Author Response
We thank you for your valuable comment. The water vapor transmission rate (WVRT) was measured by conducting ASTM F1249 test with PERMATRAN-WR Model 3/33 (Mocon Inc., USA) which is known as standard test method for WVTR through plastic film and sheeting using a pressure modulated sensor. The test method was to seal FEP film between the wet chamber and dry chamber for about 13 hours measuring moisture transmitted through the FEP film using the pressure modulated sensor. It was done at a selected temperature (38℃) and humidity (100% R.H) to avoid uncertainness. Commercially available Dupont Teflon® FEP film material property table say that 25 µm thick FEP film’s WVTR is 7.0 g/(m2∙24 h) which is higher than our test result 4.65 g/(m2∙24 h). This result may due to the difference of test method (ASTM E-96) and environments (25℃). There are also literatures that mention FEP’s WVTR [27-29]. According to our test result and existing literatures, FEP film has quite low WVTR in comparison with other plastic films but it can’t fully prevent water vapor from passing through.

Round 2
Reviewer 1 Report
AUthors have carefully applied all the changes. I would recommend the paper for publication.
Reviewer 3 Report
The authors have modified the manuscript suitably.